# Engineering human pluripotent stem cells into a functional skeletal muscle tissue

Lingjun Rao[1], Ying Qian[1], Alastair Khodabukus[1], Thomas Ribar[2] & Nenad Bursac[1]

The generation of functional skeletal muscle tissues from human pluripotent stem cells (hPSCs) has not been reported. Here, we derive induced myogenic progenitor cells (iMPCs) via transient overexpression of Pax7 in paraxial mesoderm cells differentiated from hPSCs. In 2D culture, iMPCs readily differentiate into spontaneously contracting multinucleated myotubes and a pool of satellite-like cells endogenously expressing Pax7. Under optimized 3D culture conditions, iMPCs derived from multiple hPSC lines reproducibly form functional skeletal muscle tissues (iSKM bundles) containing aligned multi-nucleated myotubes that exhibit positive force–frequency relationship and robust calcium transients in response to electrical or acetylcholine stimulation. During 1-month culture, the iSKM bundles undergo increased structural and molecular maturation, hypertrophy, and force generation. When implanted into dorsal window chamber or hindlimb muscle in immunocompromised mice, the iSKM bundles survive, progressively vascularize, and maintain functionality. iSKM bundles hold promise as a microphysiological platform for human muscle disease modeling and drug development.

[1] Department of Biomedical Engineering, Duke University, Durham, NC 27708, USA. [2] Duke iPSC Shared Resource Facility, Duke University, Durham, NC 27708, USA. Correspondence and requests for materials should be addressed to N.B. (email: nbursac@duke.edu)

Skeletal muscle is the most abundant and regenerative tissue in the human body[1], but can be functionally compromised due to genetic, metabolic, and neuromuscular diseases, including various dystrophies[2], diabetes[3], or Huntington's disease[4]. The ability to generate in vitro physiological equivalents of human skeletal muscle could offer a versatile platform for fundamental biological studies and development of new gene and drug therapies for muscle disorders. Large-scale physiological and drug screening studies would however require a readily available and expandable source of muscle progenitor cells[5] as well as 3D culture conditions leading to formation of biomimetic muscle tissues capable of electrically and chemically induced force generation.

We recently reported the first engineering of functional 3D muscle tissues ("myobundles") made from primary human myoblasts that displayed physiological force generation and calcium ($Ca^{2+}$) transients in response to electrical and biochemical stimulation[6]. These biomimetic muscle equivalents responded like native muscle to drugs that promote or decrease muscle function demonstrating the utility as a drug-screening tool. Despite the benefits of this system, primary cells have limited proliferation in vitro[7], lose their myogenic potential with serial passaging, and can be difficult to obtain from aged donors[8], or

patients with muscular diseases[9]. Furthermore, while specific culture conditions can promote maintenance of human satellite cell myogenicity[10–12], to date this has not been done for prolonged culture periods to allow generation of significant numbers of regenerative cells. Therefore, an alternative human myogenic cell source will be required to enable large-scale physiological and pharmacological engineered muscle studies in vitro and regenerative therapies in vivo.

Human pluripotent stem cells (hPSCs)[13–15] represent attractive cell sources for engineering biomimetic skeletal muscle due to unlimited proliferative potential, their ability to differentiate into myogenic cells[16–22], maintain pathological phenotypes[21,23–26], and their suitability for genome editing to study disease variants in same genetic background or correct underlying mutations. Skeletal muscle cells can be derived from hPSCs via small molecule differentiation[14,16,21,22,27,28] or direct reprogramming[17,18,20,29,30]. While small molecule differentiation can produce multinucleated myotubes, the process is relatively slow, inefficient, and does not yield a pure expandable myogenic population that is essential for large-scale drug screening or potential therapeutic applications[31]. In contrast, direct reprogramming protocols that utilize overexpression of myogenic transcription factors Pax7 or MyoD are more rapid and efficient,

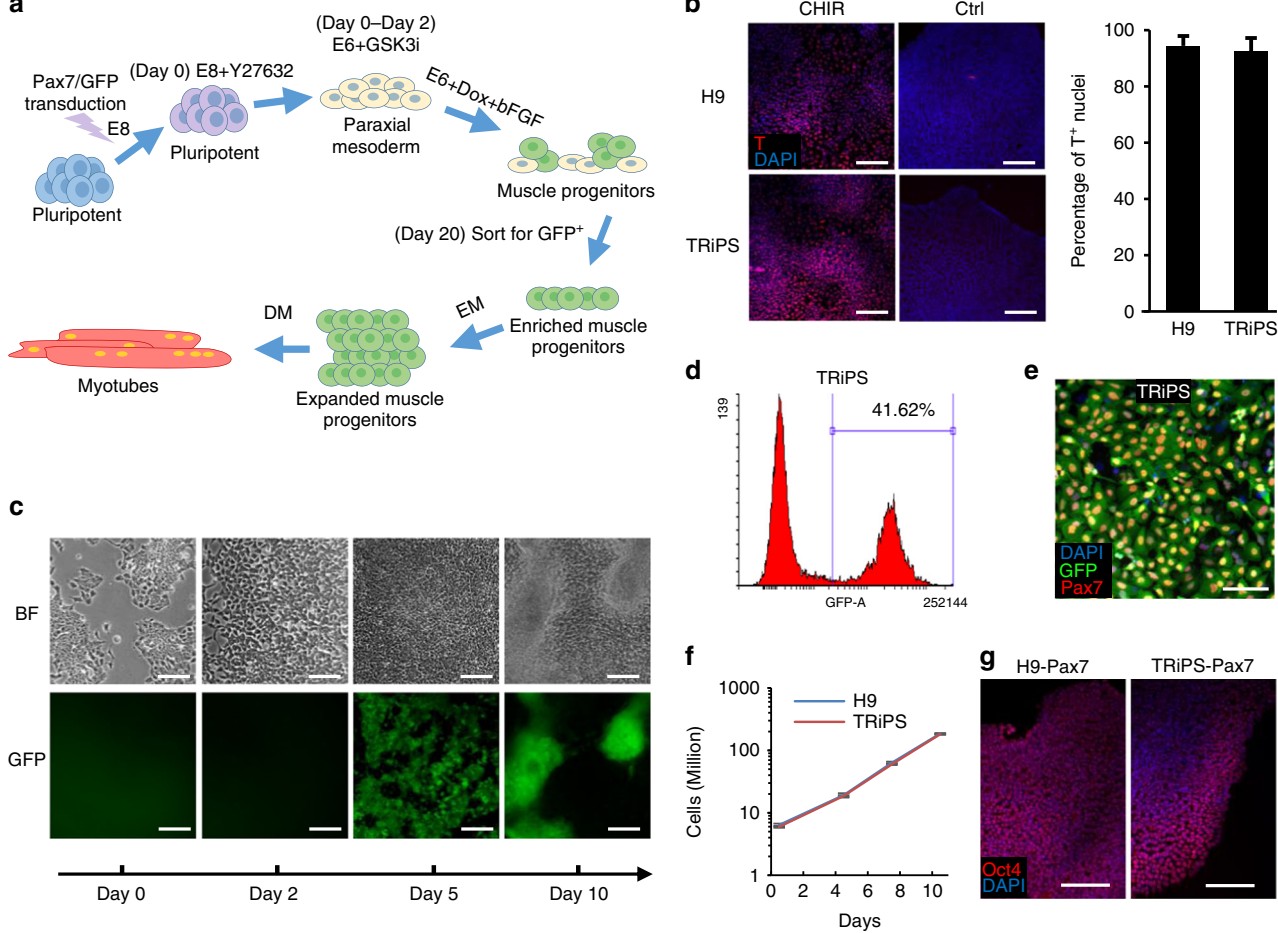

**Fig. 1** Derivation of Pax7+ myogenic progenitors from hPSCs. **a** Schematics of differentiation procedure from hPSCs to myotubes. Dox doxycycline, EM expansion media, DM differentiation media. **b** Representative immunostaining and quantification of early mesodermal marker, T, at Day 2 of differentiation. Scale bar=100 μm (n = 6 samples from 3 differentiations for each cell line). **c** Representative images showing changes in cell morphology during first 10 days of differentiation. Scale bar=100 μm. **d** Representative FACS analysis at Day 20 when GFP+ cells were sorted out for subsequent expansion. **e** Representative immunostaining of GFP and Pax7 in TRiPS derived progenitors at day 4 of expansion. Scale bar=50 μm. **f** Growth curve of hPSCs derived muscle progenitors during expansion (n = 9 samples from three expansions for each cell line). **g** Representative immunostaining of pluripotent marker Oct4 in Pax7/GFP transduced H9 and TRiPS cells after freezing and thawing. Scale bar=100 μm. Data are presented as mean ± SEM

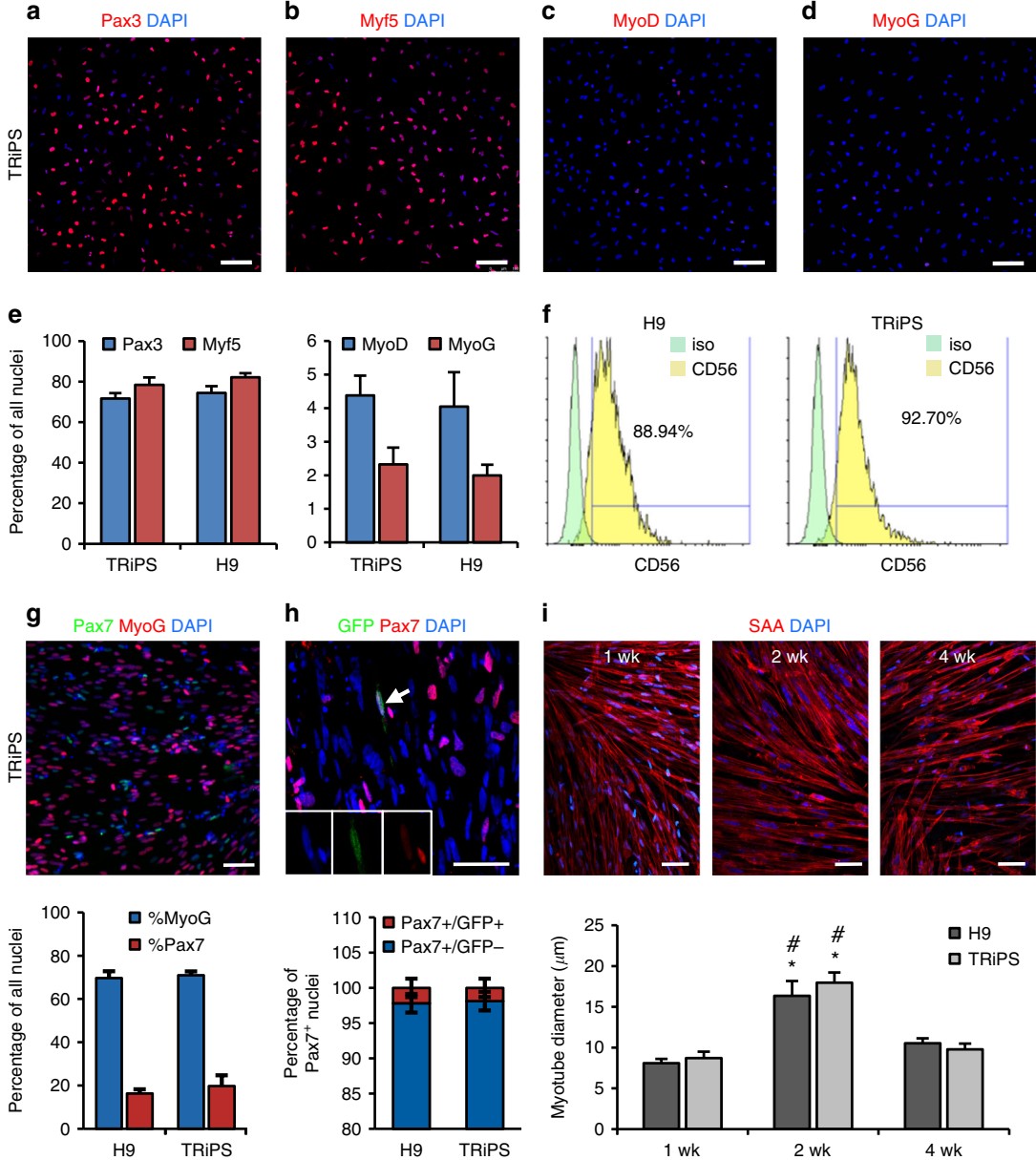

**Fig. 2** Characterization of iMPCs during monolayer differentiation. **a–e** Representative immunostaining of Pax3 (**a**), Myf5 (**b**), MyoD (**c**), and MyoG (**d**), and corresponding quantification (**e**) during iMPC expansion. Scale bar=100 μm. **f** Representative FACS analysis for CD56 in H9 and TRiPSC derived iMPCs. **g** Representative immunostaining (top) and quantification (bottom) of Pax7+ and MyoG+ cell populations for H9 and TRiPS derived myotubes at 2 weeks of monolayer differentiation. (*n* = 6 samples from 2 differentiations for each cell line). **h** Representative immunostaining and quantification of GFP+/Pax7+ and GFP−/Pax7+ cell pools at 2 weeks of monolayer differentiation. Scale bar=50 μm. (*n* = 4 samples from 2 differentiations for each cell line). **i** Representative immunostaining and quantification of myotube diameter at 1, 2, and 4 weeks of monolayer differentiation. (*$P < 0.05$ vs. 1 week, #$P < 0.05$ vs. 4 week, Tukey–Kramer HSD test; *n* = 6 samples from 2 differentiations for each cell line). Scale bars=50 μm. Data are presented as mean ± SEM

yielding >90% pure myogenic cells. Specifically, Pax7 over-expression generates a population of myogenic progenitors that can expand in vitro and populate stem cell niche when implanted into native muscle[17], suggesting their resemblance with primary satellite cells.

While several studies have shown that hPSC-derived myogenic cells can fuse with host myofibers and improve muscle function following in vivo transplantation[17,19,32,33], it is currently unknown if these cells alone can generate 3D functional skeletal muscle. Furthermore, morphological and biochemical comparisons suggest that hPSC-derived myotubes are more developmentally immature than primary myotubes, but no functional comparisons have been performed between these two cell sources.

Here, we for the first time generate functional biomimetic skeletal muscle tissues entirely from hPSC-derived myogenic cells. We first develop an efficient protocol to generate a source of expandable myogenic progenitors, termed induced myogenic progenitor cells (iMPCs) by application of the GSK3 inhibitor CHIR99021 followed by inducible expression of satellite cell marker Pax7. We derive iMPCs from both hESCs and hiPSCs and show that they differentiate into spontaneously contracting, cross-striated, multinucleated myotubes and a pool of Pax7+ cells in both 2D and 3D culture conditions. Furthermore, cylindrically shaped 3D muscle tissues ("bundles") generate active twitch and tetanic contractile forces and Ca2+ transients in response to electrical and acetylcholine stimulation and exhibit progressive

myotube hypertrophy and functional maturation during 4-week in vitro culture. Lastly, we demonstrate the ability of these engineered tissues to survive, vascularize, and maintain contractile function using two mouse implantation models. Our results suggest possibility for future use of hPSC-derived skeletal muscle tissues in physiological and pharmacological studies.

## Results

**Derivation of induced myogenic progenitor cells from hPSCs.** Two hPSC lines were used for detailed characterization of myogenic differentiation, an H9 hES cell line and a transgene-free hiPS cell line (TRiPS) from the Duke iPSC Shared Resource Facility (Supplementary Fig. 1). Undifferentiated hPSCs were transduced with lentiviruses encoding doxycycline-inducible expression of Pax7 and GFP (Supplementary Fig. 2a). In previous studies, successful commitment to paraxial mesoderm was achieved through suspension culture of embryoid bodies (EBs)[17], Wnt activation[27], or Wnt activation and BMP inhibition[28]. To achieve efficient myogenic differentiation (Fig. 1a), we first optimized induction of primitive streak and paraxial mesoderm by testing various media, and found that application of GSK3 inhibitor CHIR99021 (10 μM) in E6 medium for 2 days resulted in the highest marker expressions in both H9 and TRiPS cell lines (Fig. 1b, Supplementary Fig. 2b). For generation of myogenic progenitors, we compared two different media, E6 used previously following mesoderm induction by CHIR99021[27], and muscle induction media (MIM) used in a different Pax7 over-expression system[17]. GFP$^+$ cells appeared within 24 h of exposure to 1 μg/mL Dox and started to form 3D cell clusters after 3 days (Fig. 1c). Most of GFP$^-$ cells in MIM underwent cell death (Supplementary Fig. 2c, left) and GFP$^+$ cells failed to proliferate after sorting (Supplementary Fig. 2d, left). In contrast, the use of E6 media (Supplementary Fig. 2c and 2d, right) yielded 40–50% GFP$^+$ cells by day 20 (Fig. 1d, Supplementary Fig. 2e) allowing derivation of highly pure GFP$^+$/Pax7$^+$ cells by FACS (Fig. 1e, Supplementary Fig. 2f). Interestingly, during subsequent expansion for three passages in E6 media, percent of GFP$^+$ cells significantly decreased (Supplementary Fig. 3a, bottom and 3b), with some cells acquiring spindle shapes indicative of spontaneous differentiation despite the presence of Dox (Supplementary Fig. 3a, bottom left). We thus utilized our previously described primary myoblast media (PMM)[6], which yielded improved maintenance and proliferation of GFP$^+$ cells during expansion (Supplementary Fig. 3a, top, and b–c), as well as improved fusion and myogenesis (Supplementary Fig. 3d) after application of serum-free differentiation media (DM, Supplementary Table 1). Overall, starting with 300,000 transduced hPSCs, the optimized procedure with myogenic induction in E6 media yielded ~6 million GFP$^+$ iMPCs that could be sorted and expanded to 200 million cells after three passages in PMM media (Fig. 1f). Furthermore, the iMPCs could be cryopreserved and recovered with no effects on the GFP$^+$ population or cell viability. Without Dox induction, lentivirus-transduced hPSCs showed normal morphology and the capacity to be expanded and cryopreserved while maintaining the expression of the pluripotent marker Oct4 (Fig. 1g).

**Differentiation of iMPCs in monolayer culture.** Immunostaining of expanded iMPCs showed abundant expression of early myogenic markers Pax3 (>70%)[34] and Myf5 (~80%)[35], but not later markers MyoD (~4%) and myogenin (MyoG, ~2%) (Fig. 2a–e), with 90% cells expressing myoblast surface marker CD56 (Fig. 2f). Compared to passage-matched human primary myoblasts, iMPCs exhibited less committed myogenic state as evidenced from higher expression of PAX3 and PAX7 and lower

expression of MYF5, MYOD, and MYOG (Supplementary Fig. 4). Within 4 days in differentiation medium, iMPCs became spindle-shaped (Supplementary Fig. 3d), increased expression of MYOD and MYOG, decreased expression of GFP (Supplementary Fig. 5a–c, f–h), and attained elongated nuclear morphology typical of myonuclei (Supplementary Fig. 5c). With further differentiation, MYOD expression decreased while MYOG expression became steady (Supplementary Fig. 5d–g), thus reflecting the genetic hierarchy characteristic for muscle development and regeneration. By 2 weeks of differentiation, iMPCs readily fused into spontaneously contracting (Supplementary Video 1), multinucleated myotubes (7–10 nuclei), expressing MyoG, acetylcholine (ACh) receptors, and the muscle-specific structural protein sarcomeric α-actinin (SAA) arranged in a cross-striated pattern (Supplementary Fig. 6). Both TRiPS and H9 differentiations yielded ~70% MyoG$^+$ nuclei and a significant number of Pax7$^+$ cells (19.7 ± 4.9% and 16.3 ± 1.9%, respectively) (Fig. 2g). Over 98% of Pax7$^+$ cells were negative for GFP (Fig. 2h), indicating that they expressed nuclear Pax7 endogenously and without the need for doxycycline. In addition, these endogenously expressing Pax7$^+$ cells resided near multinucleated myotubes thus resembling a satellite-like cell pool. To assess Ca$^{2+}$ handling as an important indicator of muscle function[36], iMPCs were transduced by a lentivirus carrying a MHCK7-GCaMP6 cassette as previously described[6,37], then differentiated for 2 weeks. By recording the GCaMP6 fluorescence, we observed spontaneous Ca$^{2+}$ transients, as well as robust twitch (1 Hz) and tetanic (20 Hz) responses to electrical stimulation (Supplementary Video 1), akin to native muscle[38]. Interestingly, with longer monolayer culture, average myotube diameter first increased between 1 and 2 weeks but then decreased by week 4 (Fig. 2i), likely due to gradual detachment of larger, spontaneously contracting myotubes. Together, under optimized conditions, hPSC-derived iMPCs were capable of highly efficient 2D differentiation in vitro, yielding both the functional myotubes and resident Pax7$^+$ cells, the two main constituents of native skeletal muscle.

**Generation of 3D skeletal muscle bundles from iMPCs.** Since 2D culture could not maintain maturity of iMPC-derived myotubes for extended period, we developed a protocol for engineering functional 3D skeletal muscle (iSKM) bundles by embedding cells in fibrin-based hydrogel[6,37] (Fig. 3a). The 3D bundles were cultured for 4 days in expansion media before switching to Dox-free, serum-free differentiation media, and similar to monolayer culture, they started to spontaneously contract after 10–14 days (Supplementary Video 2). At 2 weeks of differentiation, iSKM bundles contained densely packed, aligned, cross-striated myotubes that ubiquitously expressed membrane-localized dystrophin and ACh receptors, and were embedded in a laminin and collagen I rich matrix (Fig. 3b–d, Supplementary Fig. 7). Similar to 2D culture, iSKM bundles harbored a fraction of Pax7$^+$ cells (~12%) in between MyoG$^+$ myotubes (Fig. 3e). With extended culture, myotubes in bundles improved sarcomeric structure (Fig. 3f) and increased diameter (Fig. 3g). From transmission electron micrographs (Fig. 3h–j), 4-week differentiated myotubes contained registered sarcomeres with alternating A-bands and I-bands (92% of cells), electron-dense Z-lines (92% of cells), and distinct M-lines (86% of cells) and H-zones (88% of cells) located at the center of A-bands (Fig. 3h, i, Supplementary Fig. 8a). Across individual sarcomeres, the majority exhibited distinct Z-lines (72.6%), M-lines (54.3%), and H-zones (68.2%, Supplementary Fig. 8b), and had average length (2.11 ± 0.67 μm, Supplementary Fig. 8c) characteristic of native human sarcomeres[39]. Similar to previous report[40], we identified various junctional structures between T-tubules and terminal

cisternae of sarcoplasmic reticulum (SR), including triads (Fig. 3j), inverted triads (Supplementary Fig. 9a, b), diads (Supplementary Fig. 9c), and structures containing multiple T-tubules and SR cisternae (Supplementary Fig. 9d). Furthermore, some T-SR structures were found adjacent to the A–I junctions (Fig. 3j, Supplementary Fig. 9a, c), as typical of native skeletal muscle.

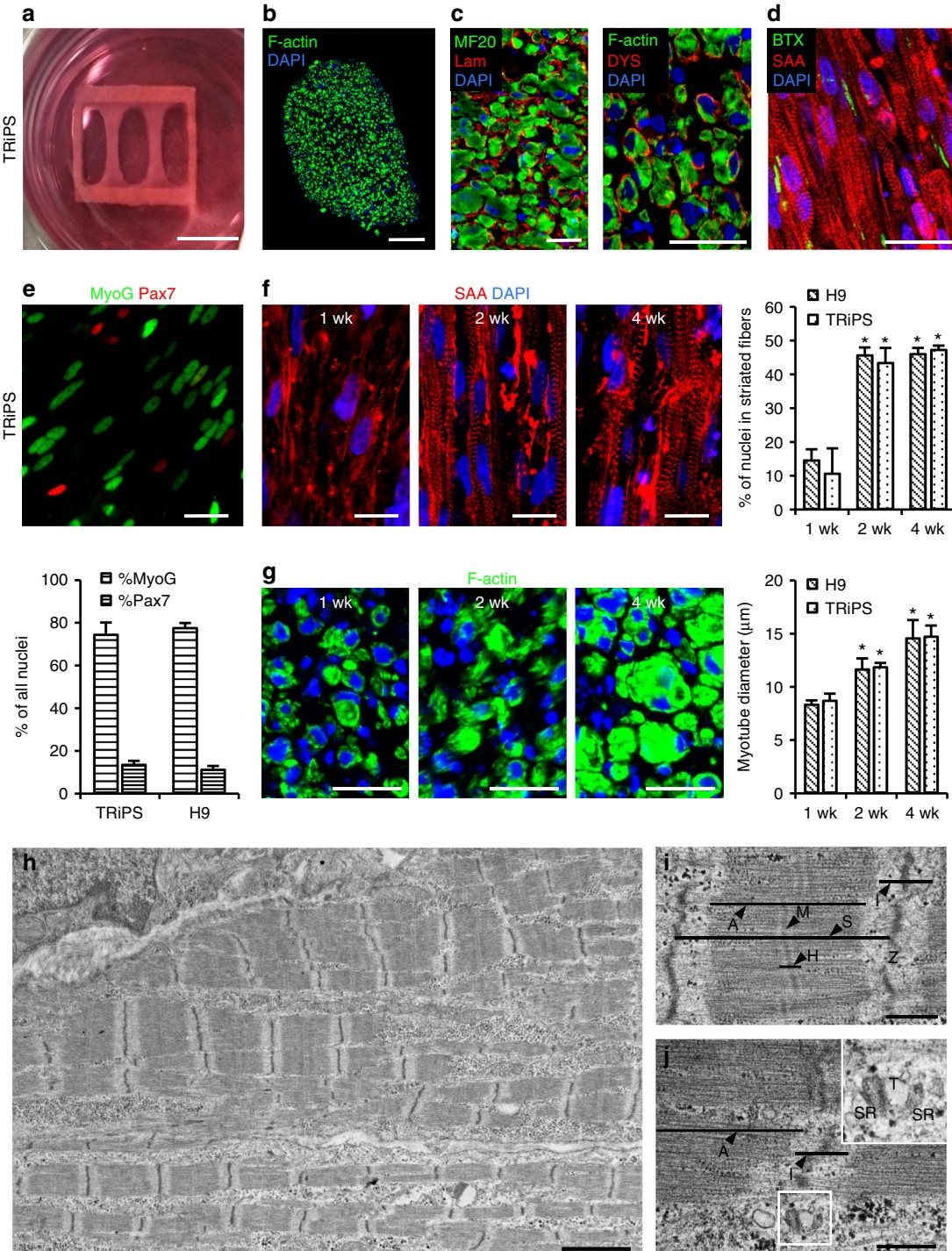

**Fig. 3** Structure of hPSC-derived iSKM bundles. **a** A 2-week differentiated iSKM bundle pair anchored within a nylon frame. Scale bar=5 mm. **b**, **c** Representative immunostaining of dense, uniformly distributed, myotubes in bundle cross-section. Scale bars: (**b**) 75 μm, (**c**) 25 μm. Lam laminin, DYS Dystrophin. **d** Representative longitudinal section of 2-week differentiated bundles showing aligned, cross-striated myotubes with α-bungarotoxin (BTX) labeled acetylcholine receptors. Scale bar=25 μm. **e** Immunostaining and quantification of Pax7[+] and MyoG[+] nuclei in 2-week H9 and TRiPS bundles ($n = 3$ bundles for each cell line). Scale bar=25 μm. **f** Immunostaining and quantification of cross-striated myotubes ($n = 3$ bundles for each cell line, *$P < 0.05$ vs. 1wk group, Tukey–Kramer HSD test). Scale bars=10 μm. **g** Immunostaining and quantification of myotube diameter ($n = 3$ bundles for each cell line, *$P < 0.05$ vs. 1wk group, Tukey–Kramer HSD test). Scale bars=25 μm. **h**, **i** Ultrastructure of iSKM bundles at 4 weeks of differentiation. A A-band, M M-line, H H-zone, S sarcomere, I I-band, Z Z-line. Scale bars=500 nm. **j** A triad junction formed close to Z-line. T T-tubule, SR sarcoplasmic reticulum. Scale bar=500 nm. Data are presented as mean ± SEM

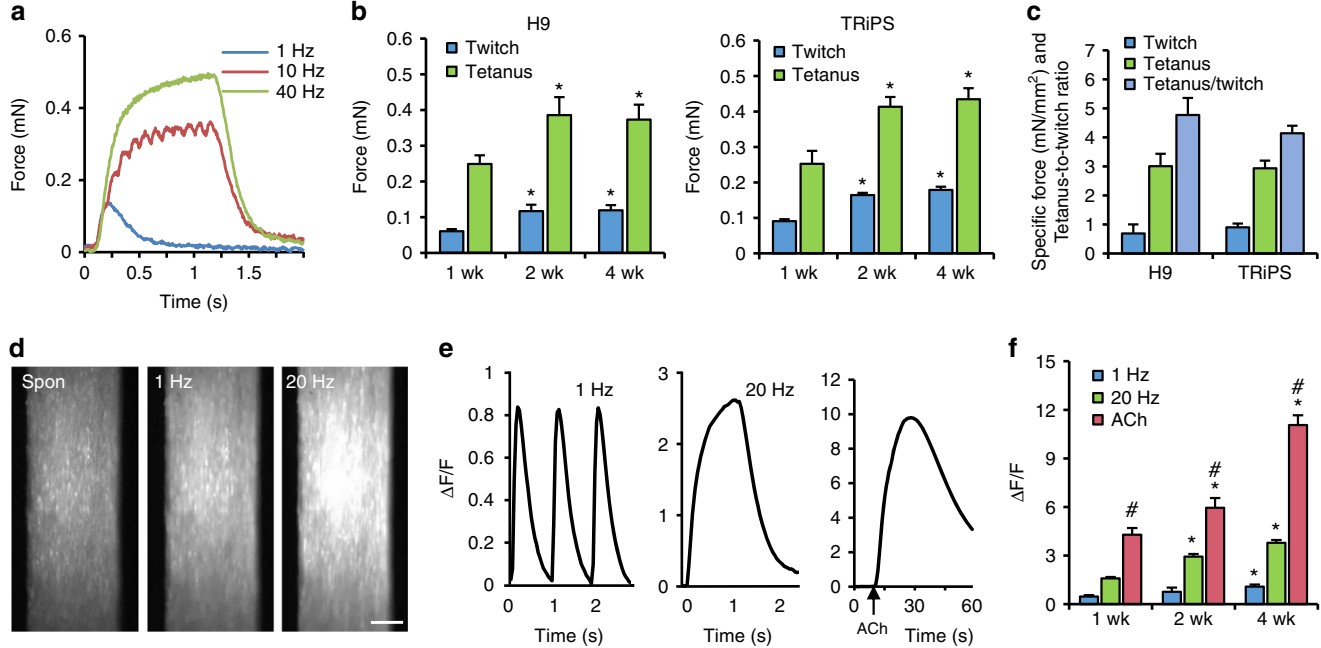

**Fig. 4** Contractile properties of hPSC-derived iSKM bundles. **a** Representative contractile force traces of a 4-week TRiPS-derived bundle shows increase in contractile force with increase in stimulation frequency and formation of tetanic contraction. **b** Increase of twitch and tetanus (at 40 Hz) force with time of culture. (*$P < 0.05$ vs. 1 week, Tukey-Kramer HSD test; $n = 6-8$ bundles per group at each time point). **c** Specific force and tetanus-to-twitch ratio of 2-week H9 and TRiPS derived bundles ($n = 10$ bundles for H9, 12 bundles for TRiPSC). **d** Representative peak GCaMP6 fluorescence images during spontaneous (spon) and electrically stimulated calcium transients in 2-week bundles. Scale bar=100 μm. **e** Representative traces of GCaMP6 signal from 1 Hz, 20 Hz, and acetylcholine (ACh, 10 mM) stimulated 2-week bundles. **f** Amplitude of electrically stimulated and ACh induced calcium transients increase with time of bundle culture. (*$P < 0.05$ vs. 1 week, #$P < 0.05$ vs. 20 Hz, Tukey–Kramer HSD test; $n = 4-8$ bundles per group at each time point). Data are presented as mean ± SEM

**Functional properties of iSKM bundles.** Generation of contractile force and calcium transients in response to electrical or chemical stimulation is a key functional signature of native and engineered skeletal muscles[41–43]. Consistent with native muscle[44] and primary human myobundles[6], iSKM bundles showed twitch and tetanic contractions, and a positive force–frequency relationship in response to electrical stimulation (Fig. 4a). For both hPSC lines, contractile force was increased from 1 week to 2 and 4 weeks of culture (Fig. 4a, b), reaching average specific force (force per cross-sectional area) of ~0.8 mN/mm$^2$ for twitch and ~3 mN/mm$^2$ for tetanus, with tetanus-to-twitch ratio ($4.8 \pm 1.6$ for H9, $4.1 \pm 2.1$ for TRiPS, Fig. 4c) being similar to those of primary myobundles and adult human muscle[6,45]. Simultaneously, passive tension and twitch kinetics of iSKM bundles remained stable during 4-week culture (Supplementary Fig. 10) and similar in magnitude to that of primary human myobundles[6] indicating similar fiber type compositions. Robustness of the methodology was further confirmed by generating functional bundles from 2 additional hPSC lines, GM25256[46] hiPSC and Fucci[47] hESC line (Supplementary Fig. 11). Consistent with contractile force results, iSKM bundles also generated robust spontaneous and electrically-stimulated Ca$^2+$ transients (Fig. 4d, Supplementary Video 2) with amplitudes that increased at higher stimulus frequencies (Fig. 4e, left and middle). Based on the presence of ACh receptors on bundle myotubes (Fig. 3d, supplementary Fig. 7d), we further tested their response to a bolus of 10 mM ACh, which resulted in a fast increase of GCaMP6 signal, indicating the release of Ca$^{2+}$, followed by a slow decrease due to the dilution of ACh in the bath (Fig. 4e, right, Supplementary Video 3). ACh-induced calcium response was larger than that induced by tetanic stimulation, with both responses showing increased amplitudes with time of 3D culture (Fig. 4f).

**Muscle gene expression in iMPCs and iSKM bundles.** We further compared the effects of 2D and 3D culture on myogenic differentiation of iMPCs by quantifying mRNA expression levels of various myosin heavy chain isoforms and Ca$^{2+}$ handling-related genes. Based on the gene profile heatmap, most myosin isoforms and Ca$^{2+}$-handling related genes exhibited higher expression in 3D versus 2D culture (Fig. 5a). In the iSKM bundles, three adult myosin heavy chain isoforms (*MYH1, MYH2, and MYH7*) showed significant upregulation during 4-week culture, while embryonic myosin heavy chain (*MYH3*) significantly increased from 1 to 2 weeks but subsequently decreased from 2 to 4 weeks of culture (Fig. 5b). This pattern of expression, indicating a shift from developmental to adult myosins, is consistent with findings during native muscle development[48], differentiation[48,49] and regeneration[50]. In contrast, *MYH3* expression in 2D cultures only increased with time of culture and no changes were found for *MYH1* and *MYH8* expression (Fig. 5b). Moreover, all Ca$^{2+}$-related genes (*CASQ2, CASQ1, SERCA1, SERCA2, NCX1, PLN, PMCA1*, and *RYR1*) were significantly upregulated at 2 and/or 4 weeks in 3D relative to 2D culture (Fig. 5c), further indicating advanced iMPCs differentiation in iSKM bundles. We then compared gene expression profiles of 2-week differentiated iSKM bundles with primary human myobundles[6] differentiated in the same media as well as native adult human muscle (Supplementary Fig. 12). Compared to native muscle, both iSKM and primary muscle bundles showed increased expression of embryonic (*MYH3*) and neonatal (*MYH8*) and decreased expression of adult (*MYH1,2,7*) myosin heavy chain isoforms and calcium-handling genes, indicating a relatively immature phenotype. Compared to primary myobundles, iSKM bundles showed higher *MYH8* and lower *MYH1* expression, indicating decreased maturation, while Ca$^{2+}$-related genes were expressed in mixed fashion, some at lower (*CASQ1, NCX1, PLN*) and some at higher (*SERCA1,*

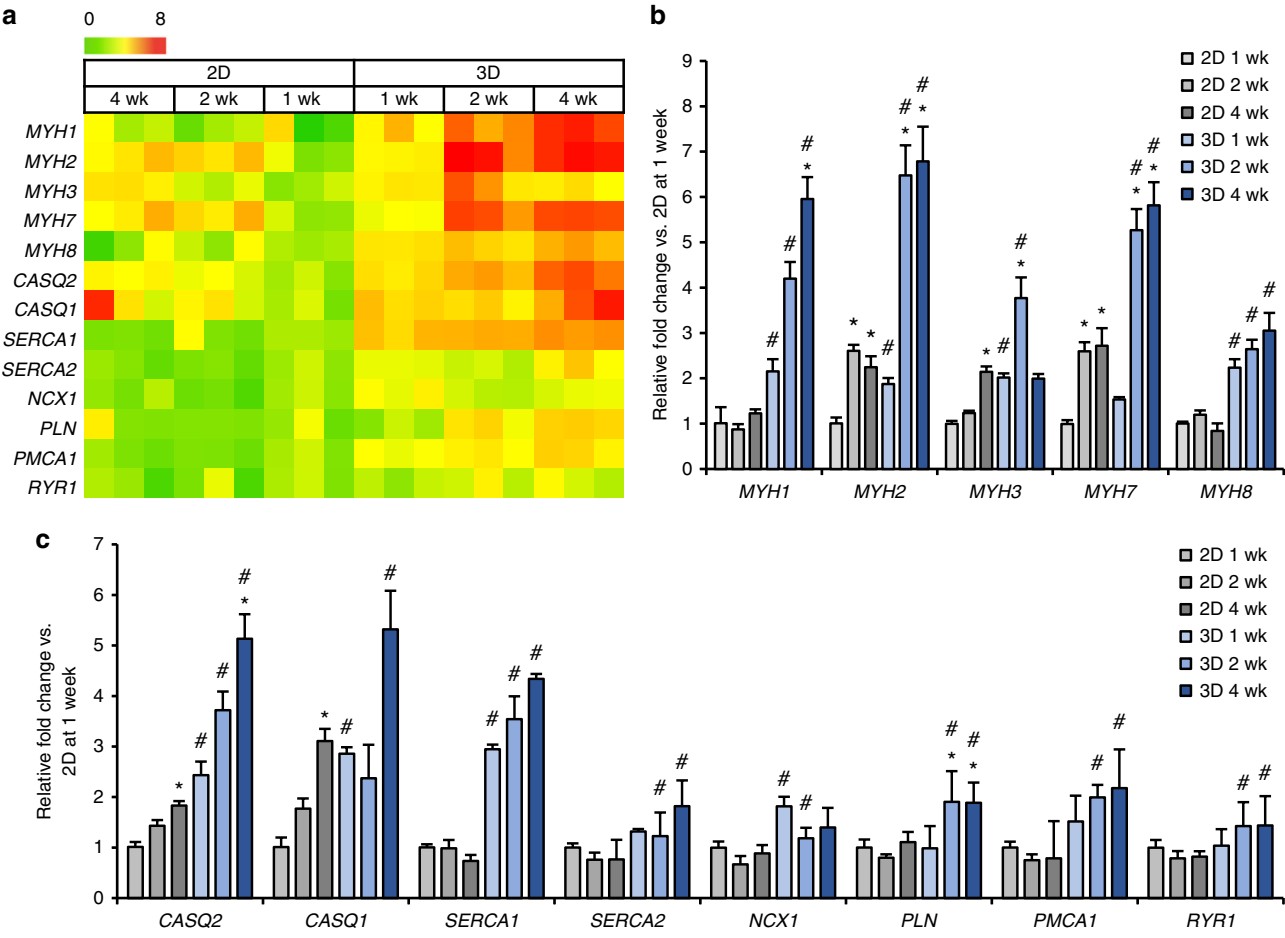

**Fig. 5** mRNA expression during 2D and 3D differentiation of hPSC-derived iMPCs. **a** Heat-map of mRNA expression in monolayers (2D) and iSKM bundles (3D) at 1, 2, and 4 weeks of differentiation, shown for three independent experiments. **b, c** qPCR results for different myosin heavy chain (MYH) isoforms (**b**) and calcium handling related genes (**c**). (*$P < 0.05$ vs. 1wk, #$P < 0.05$ vs. 2D group, Tukey–Kramer HSD test; $n = 6$–8 monolayers or bundles from three differentiations, data are presented as mean ± SEM)

PMCA1, RYR1) expression levels. Expectedly, contractile function of iSKM bundles was inferior to that of primary myobundles (twitch 1.5 ± 0.1 mN, tetanus 2.7 ± 0.2 mN; Supplementary Fig. 13), which in the presence of serum-free, N2-supplemented differentiation media exhibited improved strength compared to our original studies[6]. Together, 3D culture environment of iSKM bundles appears superior to standard 2D culture for in vitro myogenic differentiation of hPSCs; however, further optimization will be required for iSKM bundles to match the maturation state and functionality of primary human engineered muscle.

**Implantation of iSKM bundles into immunodeficient mice.** To assess the ability of the iSKM bundles to survive and function in vivo, we first implanted 2-week differentiated GCaMP6-labeled bundles into dorsal skinfold window chambers in immunocompromised mice, using our previously described methods[37] (Fig. 6a). From intravital imaging (Fig. 6b), we observed that host capillaries infiltrated the periphery of iSKM bundles by 9 days post-implantation (PI) and assembled into networks across the entire bundle by 12–15 days PI (Fig. 6b). Quantification of blood vessel density (BVD, Supplementary Fig. 14) revealed significantly increased vascularization in bundle vs. non-bundle area after 9 days PI (Fig. 6c). By 2–3 weeks PI, flattened muscle implants containing GFP+ myotubes were found to overlay native skin muscle (Fig. 6d). Consistent with intravital images, CD31+

blood vessels occupied both the periphery and interior of implanted bundles (Fig. 6e), while whole-tissue staining revealed that the implants preserved myotube alignment (Fig. 6f) and contained Pax7+ cells (Fig. 6g). Importantly, implanted bundles remained functional as evidenced by in vivo imaging of spontaneously fired GCaMP6-reported Ca²⁺ transients and ex vivo recording of twitch and tetanus responses to electrical stimulation (Fig. 6h–i, Supplementary Video 4; 18/20 bundles firing transients). Compared to age-matched in vitro cultured iSKM bundles, 15-day implanted muscle showed lower ΔF/F amplitude of GCaMP6 signal (Fig. 6i, right), which could be attributed to potential cell loss in vivo, as well as increased background fluorescence from autofluorescing host tissue or calcium-overloaded cells. We next validated the ability of 2-week differentiated iSKM bundles to survive and continually function when implanted into the tibialis anterior (TA) muscle of immunocompromised mice (Fig. 6j). To improve the signal-to-noise ratio in these studies, we labeled iMPCs with a red-fluorescing Ca²⁺ indicator R-GECO[51] instead of the green-fluorescing GCaMP6. Consistent with the window chamber results, implanted iSKM bundles survived, contained dense myotubes, and were distinguished from host muscle by specific labeling of human nuclear antigen (HNA, Fig. 6k). Moreover, ex vivo assessment at 1 week PI showed that implants exhibited physiological twitch and tetanus R-GECO-reported Ca²⁺ transients (Fig. 6l, left, Supplementary Video 5 and Fig. 15; 8/8 bundles firing transients)

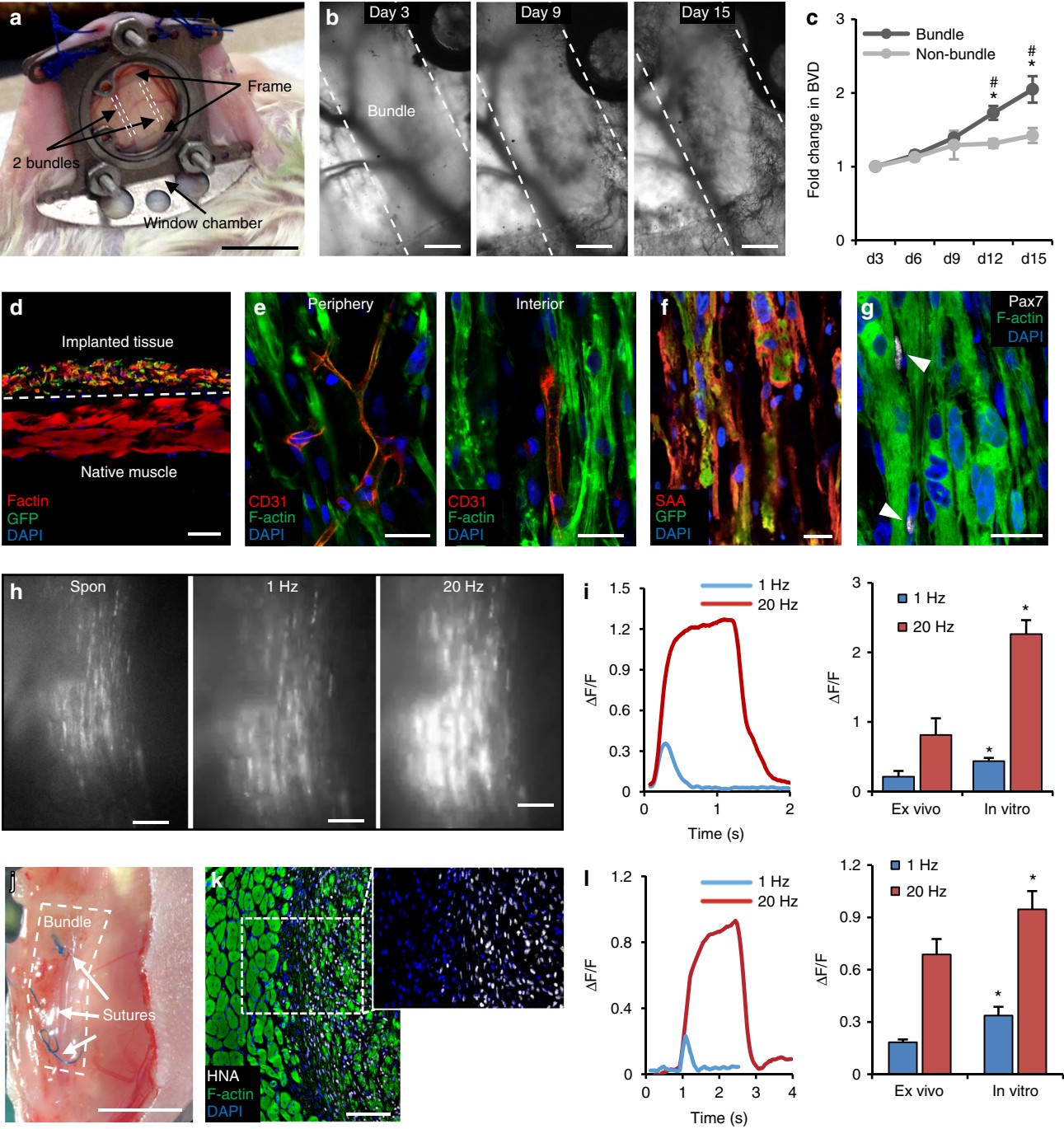

**Fig. 6** Engraftment and function of iSKM bundles implanted in immunodeficient mice. **a** Implanted bundles (delineated by dashed lines) within the dorsal skin-fold window chamber. Scale bar=5 mm. **b** Representative images showing progressive ingrowth of host blood vessels in implanted iSKM bundles at day 3, 9, and 15 post-implantation, PI. Scale bar=200 μm. **c** Relative fold change in blood vessel density (BVD) with time PI in bundle and non-bundle areas (*n* = 16 bundles (2 differentiations) from 12 mice (6-10wk); *P < 0.0001 vs. day 3, #P < 0.0001 vs. non-bundle area, Tukey–Kramer HSD test, data are presented as mean ± SEM). **d** Representative cross-section of implanted bundle with GFP-positive myofibers and underlying host skin muscle. Scale bar=50 μm. **e** Vessel organization at the periphery and interior of implanted bundle with endothelial cells labeled by CD31. Scale bar, 25 μm. **f** Representative staining of implanted bundle showing aligned myofibers positive for sacromeric α-actinin, SAA. Scale bar=25 μm. **g** Pax7[+] satellite cells (arrowheads) are found adjacent to myotubes within implanted bundles. Scale bar=25 μm. **h** Representative images of GCaMP6 signals from implanted bundles recorded ex vivo without (spontaneous) or during 1 Hz and 20 Hz electrical stimulation at 15 days PI. Scale bar=200 μm. **i** Representative traces (left) of ex vivo recorded GCaMP6 Ca[2+] transients during 1 Hz and 20 Hz stimulation of 15 day PI bundles and corresponding amplitudes (right) compared to age-matched in vitro cultured bundles (*P < 0.05 vs. ex vivo, Tukey–Kramer HSD test; *n* = 18 bundles from 10 mice for ex vivo and 6 bundles for in vitro group, data are presented as mean ± SEM). **j** iSKM bundles implanted into mouse TA muscle. Scale bar=5 mm. **k** Representative TA muscle cross-section showing 1-week implanted iSKM bundle labeled with human nuclear antigen (HNA). Scale bar=200 μm. **l** Representative traces (left) and amplitudes (right) of ex vivo recorded R-GECO Ca[2+] transients during 1 Hz and 20 Hz stimulation of iSKM bundles implanted in TA muscle and age matched in vitro controls (*P < 0.05 vs. ex vivo, Tukey–Kramer HSD test; *n* = 8 bundles from 8 mice (6–8wk) for Ex vivo and 6 bundles for in vitro group, data are presented as mean ± SEM)

with amplitudes smaller than those of age-matched in vitro controls (Fig. 6l, right). Overall, these results showed that human iSKM bundles can successfully engraft and retain functionality following implantation into skin and muscle environments in mice.

## Discussion

Our study demonstrates engineering of the first fully functional 3D skeletal muscle tissues derived from human pluripotent stem cells. Using four distinct hPSC sources, we developed a reproducible method to generate expandable myogenic progenitors, termed iMPCs, capable of highly efficient differentiation into multinucleated myotubes in 2D cell culture. When cultured in a 3D hydrogel environment, iMPCs structurally reorganize to generate aligned functional skeletal muscle tissues (iSKM bundles) that can generate twitch and tetanic contractions and $Ca^{2+}$ transients in response to electrical and neurotransmitter stimulation. Over a 4-week culture period, 3D iSKM bundles undergo progressive myotube hypertrophy and functional enhancement and attain more advanced levels of myogenic differentiation compared to age-matched 2D monolayers. Finally, we show that iSKM bundles successfully engraft and retain functionality upon implantation in two different environments in immunocompromised mice, suggesting potential utility of this platform for both in vitro and in vivo investigations.

Due to their unlimited self-renewal capacity and potential to differentiate into any cell type, hPSCs represent a promising source for the generation of skeletal muscle cells for disease modeling, drug screening, and cell therapy. For the maximum utility, hPSC differentiation protocols must enable efficient reprogramming to the lineage of choice and yield a pure and readily expandable population of progenitor cells capable of efficient terminal differentiation. To date, most of myogenic differentiation protocols do not permit subculture and expansion of highly pure myogenic progenitors[18,20,27,29]. For example, proliferative myogenic progenitors can be derived via over-expression of Pax7 in embryoid body (EB)-derived monolayers[17], however, non-specific differentiation of EBs can impede robust paraxial mesoderm commitment (Supplementary Fig. 2b), which is essential for skeletal muscle development[52,53]. In our study, efficient paraxial mesoderm induction via small-chemical inhibition of GSK3β followed by inducible overexpression of GFP tagged Pax7, GFP+ iMPC sorting, and 3-passage expansion yielded generation of $2 \times 10^8$ Pax7+ cells starting from only $3 \times 10^5$ hPSCs, enough for engineering of ~200 iSKM bundles. Overall, our protocol has been developed to efficiently and reproducibly generate a pure and expandable myogenic progenitor population able to readily differentiate into functional muscle cells in both 2D and 3D culture environments.

Skeletal muscle is a highly organized tissue comprised of densely packed, aligned multi-nucleated myofibers that possess a pool of satellite cells (SCs) residing underneath the basal lamina. hPSC-derived iMPCs switched to Dox-free differentiation media underwent myogenic program to generate both spontaneously contractile myotubes and abutting Pax7+ cells, suggesting the formation of native-like SC niches in 2D and 3D cultures. However, more advanced stages of molecular, structural, and functional differentiation were achieved only in 3D (but not 2D) culture, which supported higher expression of adult (MYH1, MYH2, and MYH7) myosin heavy chain isoforms and contractile and $Ca^{2+}$-handling genes, myotube alignment and hypertrophy, membrane-localized dystrophin expression, formation of well-developed sarcomeres and triads, positive force–frequency relationship, and generation of tetanic contractions, all characteristic of native mature muscle. Increased levels of maturation in 3D

culture such as iSKM bundles are expected to enhance predictive capability of hPSC-based systems for use in personalized disease modeling and drug development studies.

Notably, the specific tetanic forces generated by iSKM bundles (~3 mN/mm$^2$) were lower than those of native fetal muscle (~6 mN/mm$^2$) and age-matched primary human bundles (~7–12 mN/mm$^2$)[6], and significantly lower than those of adult (~84 mN/mm$^2$) human muscle[54]. Inferiority to native muscle is expected due to lack of innervation, vascularization, and mechanical loading of the bundles. Lower specific forces in iSKM than primary human bundles may have been caused in part by a more primitive state of iMPCs compared to primary myoblasts. Specifically, during expansion, the iMPCs resemble satellite cells based on their myogenic regulatory factor expression (i.e. high expression of PAX3, PAX7 and low expression of MYF5, MYOD, and MYOG) and small rounded shape (Fig. 2a–e, Supplementary Fig. 2f, Supplementary Fig. 4). In contrast, primary SCs rapidly activate in culture as evidenced from increased cell size and expression of MYOD, while, similar to iMPCs, maintain high proliferative capacity over multiple passages. The less advanced myogenic commitment of iMPCs compared to primary myoblasts (Supplementary Fig. 4) may have eventually led to the larger abundance of endogenously PAX7-expressing SC-like cells in iSKM (Figs. 2g and 3e) compared to primary bundles[6]. On the other hand, less committed iMPCs likely underwent inferior fusion and differentiation as suggested by a smaller myotube diameter measured in iSKM (~15 μm, Fig. 3g) than primary (~20 μm)[6] bundles. Although not studied, it is expected that chronic electrical and/or mechanical stimulation during culture would further promote structural and functional maturation of iSKM bundles.

Finally, we investigated the in vivo fate of iSKM bundles using two mouse implantation models. The use of dorsal window chambers allowed us to nondestructively, in real time track angiogenesis, perfusion, and calcium transients of the implanted bundles in live mice, while the leg implantation model allowed us to assess survival and functionality of iSKM bundles in a more relevant and biomechanically challenging environment of host hindlimb muscle. Despite being initially avascular, bundles within window chambers underwent robust vascularization by host blood vessels and for 3 weeks continued to fire spontaneous and electrically induced calcium transients (Supplementary Video 4), indicating the ingrown vasculature supported the survival and function of the implanted muscle. The window chamber implants maintained not only aligned myotubes but also Pax7+ cells (Fig. 6g), suggesting the potential for further growth and differentiation in vivo. Similarly, iSKM bundles implanted in TA muscle preserved structure and fired $Ca^{2+}$ transients upon electrical stimulation (Fig. 6j–l, Supplementary Video 5).

Our study is the first to demonstrate engineering of 3D contractile skeletal muscle tissues derived from human pluripotent stem cells. With prolonged time in culture, the iSKM bundles undergo structural and functional maturation and maintain Pax7+ cell pool, thus holding promise for translational use in physiological screening of novel gene and drug treatments for rare human muscle diseases. Furthermore, regarding their expression of functional acetylcholine receptors and the ability to survive, vascularize, and function in vivo, iSKM bundles might provide a foundation for the future development of hPSC-based therapies for muscle loss or dysfunction.

## Methods

**Preparation of lentivirus**. Lentiviruses were prepared as previously described[6,55]. Twenty-four hours prior to transfection, $6 \times 10^6$ 293FT cells (Life Technologies, R700-07) were seeded onto a 10 cm gelatin coated 10 cm dish. Following the manufacturer instructions, 10 μg DNA expression lentiviral plasmid (FUW-

M2rtTA, pRRL-TetO-Pax7-IRES-EGFP, pRRL-MHCK7-GCaMP6, or pRRL-MHCK7-R-GECO) was co-transfected with 5 µg psPAX2 and 2 µg VSVg by Lipofectamine2000 (Thermo). The supernatant was collected 48–72 h post transfection, concentrated with Lenti-X concentrator (Clontech) at a 3:1 ratio (supernatant: concentrator) to 1/100 of the original volume, and stored at −80 °C for later use. Lentiviruses encoding Doxycycline-inducible Pax7/GFP were used to transduce undifferentiated hPSCs, while those encoding genetically encoded calcium indicators GCaMP6[56] or R-GECO[51] were used to transduce iMPCs before formation of engineered muscle.

**Myogenic differentiation of hPSCs into iMPCs.** Human H9 (obtained from WiCell Institute), TRiPS, GM25256[46], and Fucci (gift from Dr. Stephen Dalton[47]) pluripotent stem cell lines were maintained in feeder-free conditions in E8 medium (Stemcell Technologies) and were routinely tested for Mycoplasma contamination using commercially available kits (MycoAlert, Lonza). hPSC colonies were dissociated into single cells with Accutase (Stemcell Technologies) and seeded onto Matrigel (Corning) coated 6-well plates at a cell density of $1 \times 10^3/cm^2$. Twenty-four hours post-plating, cells were infected with Tet-on lentivirus, the infected hPSCs were kept in E8 for expansion, then dissociated into single cells with Accutase and seeded onto matrigel coated 6-well plates in E8 supplemented with Y27632 (5 µM, Tocris) at $3.3 \times 10^4$ cells/cm². The following day, E8 media was replaced with E6 media and cells were cultured for 2 days supplemented with CHIR99021 (10 µM, Selleck Chemical), after which CHIR99021 was removed and E6 media supplemented with 1 µg/mL Dox (Sigma) for 18 days until GFP$^+$ induced myogenic progenitor cells (iMPCs) were sorted by FACS as described below. During differentiation of iMPCs, 10 ng/mL bFGF (R&D) was added starting at day 5 to enhance proliferation of GFP$^+$ cells.

**Flow cytometry analysis.** Cells were dissociated with 0.25% Trypsin-EDTA, counted and washed with PBS, then resuspended in flow buffer (Supplementary Table 1) at a concentration of $2 \times 10^6$ to $1 \times 10^7$ cells/mL. To count cells expressing Tra-1-81 or CD56, anti-Tra-1-81 (Stemgent, 09-0011) or anti-CD56 (PE, R&D, FAB2408P) antibodies and isotype matched controls were applied according to manufacturer's instructions and cells were analyzed using FACSCanto™ II flow cytometer (BD Biosciences) in Duke University Flow Cytometry Shared Resource. Cell population of interest was first gated for cell size and granularity, and then for the expression level of Tra-1-81 or CD56.

**Sorting of iMPCs.** At differentiation day 20, cells were dissociated with 0.25% Trypsin-EDTA (Thermo) and washed in neutralizing media (Supplementary Table 1). Detached cells were centrifuged at 300 g for 5 min, then resuspended in sorting solution (Supplementary Table 1) and filtered through 30 µM filter (SYSMEX) to remove clusters and debris. Single cell suspensions were kept on ice until sorting, with undifferentiated hPSCs used as negative control. Cells were sorted for GFP using MoFlo® Astrios™ cell sorter (Beckman Coulter) in Duke University Flow Cytometry Shared Resource.

**Expansion of iMPCs.** After sorting, GFP$^+$ iMPCs were kept on ice in collecting solution (Supplementary Table 1), spun down at 300 g for 5 min, and resuspended in fresh E6 media supplemented with Y27632, Dox, and bFGF, then seeded at $4 \times 10^4/cm^2$ in Matrigel-coated flasks. After 24–48 h of post sorting, cells were incubated in expansion media (EM, Supplementary Table 1), supplemented with Dox and bFGF, and passaged at a 1:3-1:6 ratio every 3-4 days after reaching 80% confluence.

**2D differentiation of iMPCs.** iMPCs were seeded at the density of $1 \times 10^5/cm^2$ on Matrigel-coated dishes and after reaching 100% confluence, EM was washed out with PBS and switched to differentiation media (DM, Supplementary Table 1) that was changed every other day.

**Fabrication and differentiation of iSKM bundles.** Three-dimensional engineered muscle tissues (iSKM bundles) were formed within polydimethylsiloxane (PDMS) molds containing two semi-cylindrical wells (7 mm long, 2 mm diameter), cast from 3D-machined Teflon masters, similar to our previously described methods[6,37]. PDMS molds were coated with 0.2% (w/v) pluronic (Invitrogen) for 1 h at room temperature to prevent hydrogel adhesion. Laser-cut Cerex® frames (9 × 9 mm², 1 mm wide rim) positioned around the 2 wells served to anchor bundle ends and facilitate handling and implantation. Cell/hydrogel mixture (Supplementary Table 1) was injected into the PDMS wells and polymerized at 37 °C for 30 min. Formed iSKM bundles were kept on rocking platform in EM supplemented with 1 µg/mL Dox and 1.5 mg/mL 6-aminocaproic acid (ACA, Sigma) for 4 days. Media was then switched to DM supplemented with 2 mg/mL ACA and 50 µg/mL ascorbic acid (Sigma), with media changed daily.

**Engineering of primary human myobundles.** Native human skeletal muscle samples were obtained through standard needle biopsy or surgical waste from donors with informed consent under Duke University IRB approved protocols (Pro00048509 and Pro00012628). Muscle samples were minced and digested with

0.05% trypsin for 30 min at 37 °C. Isolated cells were centrifuged to remove residual enzyme and resuspended in PMM, then preplated for 2 h to reduce fibroblast fraction. After pre-plating, cells were seeded onto to a Matrigel (BD Biosciences) coated flask and expanded by passaging upon reaching 70% confluence. At passage 3 or 4, cells were detached from the flask and used to fabricate primary myobundles as described for iSKM bundles.

**Force measurement.** Contractile and passive force generation in iSKM bundles was assessed using a custom force measurement set-up as previously described[6,57,58]. Briefly, single iSKM bundles were transferred attached to frame in the bath with DM equilibrated at 37 °C. One end of the bundle was pinned to a fixed PDMS block and the other end was attached to a PDMS float connected with force transducer mounted on a motorized linear actuator (ThorLabs, Newton, NJ). The sides of the frame were cut to allow isometric measurement of contractile force and stretching of the bundles by the actuator. To assess the force–length relationship, iSKM bundle was stretched in 5% steps, then stimulated with a 40 V/cm, 10 ms long electrical pulse using a pair of platinum electrodes and the twitch force was recorded. At 20% stretch, 1 s long stimulations at 5, 10, 20 and 40 Hz were applied and the contractile force was recorded to assess the force–frequency relationship. Contractile force traces were analyzed for peak twitch or tetanus force, passive tension, time to peak twitch, and half relaxation time using a custom MATLAB program.

**Immunofluorescence.** Cells cultured in monolayers were fixed in 4% paraformaldehyde in PBS for 15 min at room temperature and iSKM bundles were fixed in 2% paraformaldehyde in PBS overnight at 4 °C while rocking. Following fixation, samples were washed twice with PBS, and kept in PBS at 4 °C for up to 1 week. Before staining, samples were blocked in PBS with 5% chick serum and 0.5% Triton-X 100. The following primary antibodies were used for immunostaining of: Oct4 (Millipore, MAB4401, 1:200), Tra-1-81 (Stemgent, 09-0011, 1:100), T (R&D, AF2085, 1:200), Pax3 (R&D, MAB2457, 1:200), Myf5 (SCBT, sc-302, 1:100), MF20 (DSHB, 1:300), sarcomeric α-actinin (SAA, Sigma, a7811, 1:200), GFP (Thermo, A6455, 1:300), laminin (Abcam, Cambridge, MA, ab11575, 1:200), Dystrophin (Abcam, Cambridge, MA, ab15277, 1:100), CD31 (Abcam, Cambridge, MA, ab28364, 1:50), MyoD (BD Biosciences, 554130, 1:100), MyoG (SCBT, sc-576, 1:100), and Pax7 (DSHB, 1:100). Corresponding fluorescently labeled secondary antibodies (1:500), α-bungarotoxin (B13422, 1:200), and phalloidin (O7466, 1:300) (all from Thermo) were applied in blocking solution (Supplementary Table 1) for 1 h. Images were acquired using a Leica SP5 inverted confocal microscope and analyzed using LSM Image Software.

**Transmission electron microscopy.** iSKM bundles at 4 weeks of culture were fixed for 30 min in 0.1 M phosphate buffer (PB) containing 2% glutaraldehyde, then postfixed with 2% OsO₄ in PB. Samples were then sequentially dehydrated in 30, 50, 70, 95, 100% acetone, kept in acetone: epoxy (1: 1) overnight, and embedded in 100% epoxy. Ultrathin sections were collected on grids and stained with uranyl acetate before examination with a Philips CM12 transmission electron microscope operated at 120 kV. An XR60 camera system (Advanced Microscopy Techniques) was used for image acquisition.

**Quantitative RT–PCR.** Native human skeletal muscle samples were obtained through standard needle biopsy or surgical waste from donors with informed consent under Duke University IRB approved protocols (Pro00048509 and Pro00012628). Total RNA from 2D cells, 3D engineered muscle bundles, and native human muscle was isolated using either RNeasy Plus Mini Kit (QIAGEN) or Aurum Total RNA Mini Kit (Bio-Rad), then reverse-transcribed by iScript cDNA Synthesis Kit (Bio-Rad). Quantitative RT-PCR for muscle related genes was performed with iTaq Universal SYBR Green Supermix (Bio-Rad) according to manufacturer's instructions. Primer information can be found in Supplementary Table 2.

**Implantation of iSKM bundles in dorsal window chambers.** All animal experiments were approved by the Duke University IACUC. NSG or nude mice (male, 6–10wk of age; 28–34 g) were anesthetized by intraperitoneal (IP) injection of ketamine (100 mg/kg) and xylazine (10 mg/kg). After removing the hair on the back, the dorsal skin was attached to a temporary "C-frame" at the center of the back. The skin was perforated in three locations to accommodate the screws of the chamber, and a circular region (~12 mm) of the forward-facing skin (i.e., cutis, subcutis, retractor and panniculus carnosis muscles, and associated fascia) was dissected away to accommodate the chamber. The forward and rearward pieces of the titanium dorsal skinfold chamber were assembled together from opposite sides of the skin, and a Cerex® frame with double-bundle constructs was laid perpendicular (verified under microscope) to the intact panniculus carnosis muscle of the rearward-facing skin, providing a source of microvessels for vascularization. A sterile cover glass was placed over the window and engineered tissue while superfusing with sterile saline solution. The chamber was then secured with suture and the "C-frame" was removed. Post-operatively, the mouse was injected subcutaneously with buprenorphine (1 mg/kg) painkiller and let to recover on a heating pad.

**Implantation of iSKM bundles into hindlimb muscle**. Adult NSG mice (male, 6–8wk of age, 25–35 g), were anesthetized by IP injection of Ketamine (100 mg/kg) and Xylazine (5 mg/kg). The hindlimb hair was clipped, skin was decontaminated with 0.5% chlorhexidine/70% ethanol, and a 1.5 cm skin incision was made parallel to the tibia. Micro-dissecting forceps was used to separate tibialis anterior (TA) muscle along a 1 cm-long midline and iSKM bundle was inserted in the muscle separation. Bundle ends and mid-point were sutured to the TA muscle, the skin was closed, and buprenorphine (1 mg/kg) was applied for post-operative pain management.

**Intravital imaging of blood vessels**. Intravital recordings in dorsal window chambers were performed in anesthetized mice on days 3, 6, 9, and 15 post-implantation (PI), as previously described[37]. Mice were anesthetized by nose cone inhalation of isoflurane and positioned on a heating pad under a microscope objective. Hyperspectral brightfield image sequences (10 nm increments from 500 to 600 nm) were captured at ×2.5 and ×5 magnification using a tunable filter (Cambridge Research & Instrumentation, Inc.) and a DVC camera (ThorLabs), as previously described[59], then converted to hemoglobin saturation maps, and processed to binary blood vessel density images (Supplementary Fig. 14) for quantification of blood vessel density (BVD).

**Intravital recording of spontaneous $Ca^{2+}$ transients**. Intravital recording of spontaneous $Ca^{2+}$ transients in dorsal window chambers was performed immediately after vessel imaging with mice still anesthetized. GCaMP6 signals in implanted bundles were recorded through a FITC-filter using a fast fluorescent camera (Andor; at 16 μm spatial and 20 ms temporal resolution).

**In vitro and ex vivo measurements of $Ca^{2+}$ transients**. Electrically-stimulated $Ca^{2+}$ transients were recorded from engineered muscle bundles after 1, 2, 4 week of in vitro culture and from muscle explants 7–15 days post implantation, as previously described[6,37]. In vitro cultured iSKM bundles and excised dorsal skins and TA muscles with engrafted bundle implants were transferred into a custom chamber mounted on an upright fluorescence microscope (Leica M165 FC, for TA muscle explants) or an inverted fluorescence microscope (Nikon TE2000-U, for window chamber explants), placed in 37 °C differentiation media (DM, Supplementary Table 1), and electrically stimulated (10 ms pulse, 3 V/mm). Resulting GCaMP6 (510–560 nm bandpass emission filter) or R-GECO (590-660 nm bandpass emission filter) signals were recorded using a fast EMCCD camera (Andor iXon 860; 24 μm spatial and 20 ms temporal resolution). Amplitudes of $Ca^{2+}$ transients were determined using the Solis software (Andor) by averaging relative fluorescence intensity (ΔF/F) from each bundle.

**Statistical analysis**. Experimental data are reported as mean ± SEM. Statistical significances were evaluated by one-way ANOVA with Tukey–Kramer HSD test using JMP Pro software. $P$ value < 0.05 was considered statistically significant. Sample sizes for in vitro experiments were determined based on variance of previously reported measurements[6]. Sample sizes for implantation experiments were determined, in part, based on cost and animal availability. No randomization of animal groups was done. No blinding of animal experiments was done.

**Data availability**. All data supporting the results of these studies are available within the paper and Supplemental Materials.

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

## Acknowledgements

We thank L. Li and S. Okuwa for assistance with virus production; M. Cook, L. Martinek, and B. Li for sorting of Pax7+ cells; N. Medvitz for preparation of TEM samples; and Dr. M. Juhas for critical reading of the manuscript. This work was supported by National Institutes of Health grants UH3-TR000505, UG3-TR002142, AR065873, and AR070543.

## Author contributions

L.R.: Experimental conception and design, acquisition of data, analysis and interpretation of data, manuscript drafting and revising. Y.Q.: Experimental conception and design, acquisition of data. A.K.: Collection and assembly of data, data interpretation, manuscript drafting and revising. T.R.: Collection and assembly of data. N.B.: Experimental conception and design, financial support, administrative support, manuscript drafting and revising, final approval of manuscript.

## Additional information

**Competing interests:** The authors declare no competing financial interests.

