## [Peer Review File · Nature Communications]

Reviewers' comments:

Reviewer #1 (Remarks to the Author):

In Rao et al, the authors engineered functional skeletal muscle bundles starting from either human embryonic or induced pluripotent stem cells. This includes establishing procedures for differentiating myogenic cells from human pluripotent cells (based largely on existing protocols), differentiating and characterizing myogenic cells into myotubes in 2-D culture, engineering 3-D muscle bundles (using their previously published approach) from the myogenic cells, performing structural, functional, and genetic characterization of the 3-D bundles, and implanting 3-D bundles into mice to demonstrate vascularization and survival in vivo. This is an impressive and thorough volume of work and the combination of human stem cell-derived myoblasts with 3-D tissue engineering is an important advance for the field.

Major Comments

In some instances, the authors refer to Pax7+ cells as satellite cells (line 156) and claim that they re-built the satellite cell niche (line 280). However, myoblasts also express Pax7, so the authors should re-word these statements, as there is insufficient evidence that these are satellite cells.

In Figure 3j, the authors claim that triad structures formed in their muscle bundles. What is the criteria that the authors used to identify these as triad structures? The morphology does not appear distinct enough to make this claim.

One limitation of this manuscript is the lack of direct comparisons of their stem cell-derived induced myogenic progenitor cells (iMPCs) to primary human myoblasts, and/or their iMPC-derived myotubes to primary human myotubes. For example, in Figure 5, the authors compare gene expression in 2-D and 3-D myotubes differentiated from iMPCs. However, a highly informative control for this experiment would be myotubes differentiated from primary skeletal myoblasts over the same time-scale to demonstrate how gene expression in iMPC-derived myotubes compares to primary myotubes.

When comparing the calcium data in Fig. 6h to Fig. 4e-f, the amplitude is much lower in the constructs explanted from the mice compared to those maintained in vitro. This is not too surprising, as the bundles likely do not survive as well in the mouse, but this difference should be addressed.

Minor Comments

In several instances, the references to supplemental figures are mislabeled. For example, line 111 should refer to Supplementary Figure 2a.

Reviewer #2 (Remarks to the Author):

The manuscript by Rao and colleagues describes how they established methods to differentiate human pluripotent stem cells (hPSCs) into functional skeletal muscle tissues (iSKM bundles) containing spontaneously contracting multinucleated myotubes and a pool of satellite cells endogenously expressing Pax7. This methods holds the promise for translational use in physiological screening of novel gene and drug treatments for rare human muscle diseases. Moreover, the authors suggest that the in vitro generated iSKM bundles hold promise as a platform for the future development of hPSC-based tissue transplantation therapies for muscle loss or dysfunction. The latter conclusion is not yet supported by their data, and experimental evidence for this claim would significantly improve this manuscript.

Major comments:

- 1) The authors claim that their method to generate iSKM bundles can be important for regeneration and therapy, but do not show this. As said, adding experimental evidence that would prove this point would significantly improve this manuscript.
- 2) The authors transplant the iSKM bundles into a dorsal skinfold window chamber. It is not entirely clear why the authors have used this approach, since the presented data only shows short-term intravital microscopy experiments and not multiple day experiments. Moreover, if the authors only want to show the spontaneous firing of Ca²⁺ transients, it seems more relevant to transplant the iSKM bundles in (injured) muscles for example in the leg and surgically expose the muscle for intravital microscopy.
- 3) The description of the transplantation and intravital experiments lack all details. How many mice have been imaged, how many spontaneous firing of Ca²⁺ transients have been observed, etc etc. This is important to judge what the efficiency is of functional fibers upon transplantation.
- 4) The authors have imaged the iSKM on day 7, 14, and 21 post-implantation. Can the authors describe whether the transplanted tissue changes over time? Can the authors show images of the same transplanted tissue over multiple days?

Minor comments:

- 1) In video 4, a spontaneous firing of Ca²⁺ transient should be visible. Can the authors show with an arrow in the movie where the firing cells is?

Reviewer #3 (Remarks to the Author):

General comments

Authors have already reported bioengineered human myobundles using human primary myoblasts (2014, 2015). This time, Authors used human ES cells (1 clone) and iPS cells (1 clone) instead of primary myoblasts. The 3D myobundles seem better than already reported 2D hiPS cells-derived skeletal muscle for in vitro models or biomaterials for medicine, but the protocol is a collection and modification of already reported protocols (GSK3i, PAX7 overexpression, 3D bundle, Window chamber etc.).

Major arguments

1. Figure 5. It is more informative to present the expression level of each gene in mature muscle tissue for comparison.
2. In the text, Authors should mention that specific force of human or mouse myofibers are more than 100 mN/mm².

Minor points

1. Which sub-unit of AChR was examined?
2. In Supplementary Figure 1, Tra-1-81 was not expressed on the surface of the cells.

Reviewer #4 (Remarks to the Author):

In their article, Rao and colleagues have made impressive strides in the engineering of functional skeletal muscle from pluripotent stem cells. They specifically show that 3D culture techniques offer substantial improvement of differentiation over 2D techniques, and that produced skeletal muscles can be successfully implanted in vivo.

While these results are exciting, I feel that further quantification is required to document the extent of differentiation, and that comparison with adult skeletal muscle is vital. Specifically, at several points of the manuscript nice example images are shown, without giving the reader an

understanding of how reproducible the findings are across each muscle and between muscles. For example, when it comes to muscle structure, the authors point out the appearance of z lines, a-band, etc. How much of the muscle had this type of appearance? The z-lines visible in Figure 3h appears to be a bit wobbly. Was this a general finding? Were t-tubules present at regular interval and density, and were the dimensions of these similar to those seen in adult muscle? The authors present force data with a comparison to literature values, but these experiments should be done in the present article to ensure that experimental conditions are comparable. In implanted muscles, how variable is the capillary network and characteristics of the Ca transient? How do these compare with native muscle?

INTRODUCTION TO REVISED MANUSCRIPT

We thank all reviewers for their highly constructive comments that we addressed in detail with additional experiments and analyses, resulting in:

- 1) 8 new or revised panels in original main figures
- 2) 3 new or revised panels in original supplementary figures
- 3) 8 new supplementary figures
- 4) 1 new supplementary video

With these revisions, we hope that reviewers will find our manuscript significantly improved and worthy of acceptance to *Nature Communications*.

Reviewer #1: We thank the reviewer for the very helpful comments that we have answered in detail. Reviewer's comments are cited in bold. The answers are in normal font. Any citations of the main or supplementary text are shown in quotes and italics. Other minor changes in the text are made to improve readability and satisfy formatting requirements of Nature Communications.

Major Comments:

1. In some instances, the authors refer to Pax7+ cells as satellite cells (line 156) and claim that they re-built the satellite cell niche (line 280). However, myoblasts also express Pax7, so the authors should re-word these statements, as there is insufficient evidence that these are satellite cells.

We thank the reviewer for this comment and agree that hPSC-derived Pax7⁺ cells may not be true satellite cells. However, the MRF protein expression in iMPCs (Fig. 2a-e) and comparison of MRF gene expression profiles between iMPCs and primary myoblasts (new Supplementary Fig. 4) show that early MRF factors Pax3 and Pax7 (known to be more abundantly expressed in quiescent satellite cells than myoblasts) are expressed at higher levels in iMPCs than passage-matched primary human myoblasts (Supplementary Fig. 4). Furthermore, Myf5, MyoD and Myogenin, markers of a more committed myogenic state, are expressed at lower levels in iMPCs than primary myoblasts (Supplementary Fig. 4). Importantly, at the protein level, only ~4% of the cells express detectable MyoD protein (Fig. 2c and e). Collectively, the MRF expression profile of Pax7⁺ cells and residing in the vicinity myotubes (Fig. 6g) suggest their resemblance to satellite cells. We have thus referred to these cells as “resembling a satellite-like cell pool” in line 159, “native-like SC niches” in line 316, “resemble satellite cells” in line 332, and “satellite-like cells” in line 29.

2. In Figure. 3j, the authors claim that triad structures formed in their muscle bundles. What is the criteria that the authors used to identify these as triad structures? The morphology does not appear distinct enough to make this claim.

We used the standard definition of a triad structure being one T-tubule tightly associated with two SR cisternae. In our TEM images, T-tubules are shown as electrolucent lumens and SR cisternae contain electron-dense granular material. Similar structures in hPSC-derived myotubes have been reported before (Ref. 37). In addition to triad structures, we observed other types of T-SR appositions including dyads (one T-tubule with one SR), inverted triads (two T-tubules with one SR), and multi-junction structure among several T-tubule and SRs. Interestingly, some of the T-SR structures in our tissues were located close to A-I junction resembling arrangement found in native muscle, and previously not reported in hPSC-derived myotubes. To address this comment, we now provide more TEM images of observed T-SR structures (revised Supplementary Fig. 9) and describe these findings in lines 182-194 of the revised Results section.

3. One limitation of this manuscript is the lack of direct comparisons of their stem cell-derived induced myogenic progenitor cells (iMPCs) to primary human myoblasts, and/or their iMPC-derived myotubes to primary human myotubes. For example, in Figure 5, the authors compare gene expression in 2-D and 3-D myotubes differentiated from iMPCs. However, a highly informative control for this experiment would be myotubes differentiated from primary skeletal myoblasts over the same time-scale to demonstrate how gene expression in iMPC-derived myotubes compares to primary myotubes.

We thank the reviewer for this very important comment. We have compared mRNA expression levels of Pax7, Pax3, Myf5, MyoD and myogenin between hPSC-derived muscle progenitors and passage-matched primary human myoblasts (new Supplementary Fig. 4). As described in our response to comment 1, we find that iMPCs have a less committed and differentiated phenotype compared to primary myoblasts. We have also compared expression levels of calcium-handling and myosin heavy chain (MYH) isoform genes among native adult human skeletal muscle, 2-week differentiated engineered primary muscle bundles (myobundles, described previously in our Ref. 5), and 2-week differentiated iSKM bundles (new Supplementary Fig. 12). We found that all calcium-handling genes showed lower expression in primary and iSKM bundles compared to native muscle, with some genes being lowest expressed in primary bundles and others in iSKM bundles. Additionally, the embryonic (MYH3) and neonatal (MYH8) MYH isoforms were increased in primary and iSKM bundles compared to native muscle, with highest levels of MYH8 found in iSKM bundles – indicating the lowest maturation state. For the adult MYH isoforms, both primary and iSKM bundles had lower expression compared to native muscle, with MYH1 (myosin IIx) being lowest expressed in iSKM bundles. Overall, this data shows that both 2wk primary and 2wk iSKM bundles are less mature than healthy adult native muscle. Furthermore, the iSKM bundles appear to be less mature compared to age-matched primary myobundles based on the expression of MYH8 and MYH1. However, the similar expression levels of myosin I and IIa (MYH7 and MYH2) in primary and iSKM bundles indicate that some maturation aspects are comparable between these engineered muscle tissues.

These findings are now described in the revised Results section in lines 143-146 and 233-248.

4. When comparing the calcium data in Fig. 6h to Fig. 4e-f, the amplitude is much lower in the constructs explanted from the mice compared to those maintained in vitro. This is not too surprising, as the bundles likely do not survive as well in the mouse, but this difference should be addressed.

We thank the reviewer for this comment. From our experience, $\Delta F/F$ values can be most reliably compared temporally within the same experiment (e.g. same batch of bundles compared at 1, 2, 4 wks of culture) and/or measurement (e.g. same bundle during twitch and tetanic contraction, or pre- and post-drug). As highlighted by the reviewer, in the original manuscript there is a significant decrease in $\Delta F/F$ in Fig. 6h compared to Fig. 4f. This difference may in part be due to a cell loss or injury after implantation, however, other contributing factors could be the differences in GCaMP6 transduction efficiency between different batches of cells as well as specific conditions of optical recording. In particular, compared to iSKM bundles assessed during *in vitro* culture, implanted bundles assessed *ex vivo* have higher background fluorescence (F), probably due to additional autofluorescence of infiltrated host cells and surrounding host tissue (note that implanted bundles remain in/on the host tissue during *ex vivo* recordings), while ΔF of implanted bundles is lower due to tissue flattening (Fig. 6d) which yields less cell layers contributing to recorded signal.

To carefully address reviewer's question, we performed additional implantation experiments (using two genetic calcium sensors with different wavelengths, green GCaMP6¹ and red R-GECO²) in both window chambers and native muscle while continuing to culture iSKM bundles from the same batch of transduced cells to obtain age-matched *in vitro* controls. We found that $\Delta F/F$ amplitudes of implanted bundles recorded *ex vivo* are 35-70% of those recorded in age-matched *in vitro* cultured bundles. These results are now shown in revised Figs. 6i and 6l and are described in lines 265-279.

Minor Comments:

1. In several instances, the references to supplemental figures are mislabeled. For example, line 111 should refer to Supplementary Figure 2a.

We thank the reviewer for noticing this omission which is now corrected in the revised manuscript.

Reviewer #2: We thank the reviewer for the very helpful comments that have significantly improved the manuscript. Reviewer's comments are cited in bold. The answers are in normal font. Any citations of the main or supplementary text are shown in quotes and italics. Other minor changes in the text are made to improve readability and satisfy formatting requirements of *Nature Communications*.

Major comments:

1. The authors claim that their method to generate iSKM bundles can be important for regeneration and therapy, but do not show this. As said, adding experimental evidence that would prove this point would significantly improve this manuscript.

We thank the reviewer for this important comment and apologize that our intention was not clearly conveyed. The aim and scope of our original and revised manuscript were to: 1) present the first-time engineering of functional 3D muscle tissues derived from hPSCs, 2) compare these tissues with engineered muscle made from primary human myoblasts, and 3) show that *in vitro* engineered hPSC-muscle can survive and continue to function upon implantation in immunocompromised mice. We agree with the reviewer that suggesting future regenerative therapy applications with current iSKM bundles is premature given that they produce sub-mN forces. For potential use in the repair of small human muscles (those of eye, lips, sphincters, etc.), the iSKM bundles will require extensive optimization to further increase force generation and tissue maturation. Therefore, we believe that such optimization and validation for regenerative therapy applications is beyond the scope of the current manuscript, though it will certainly be a primary focus in our future work. To address the reviewer's comment, we retained the discussion on potential utility of iSKM bundles in physiological and pharmacological studies, but removed mentions of potential regenerative therapy applications.

2. The authors transplant the iSKM bundles into a dorsal skinfold window chamber. It is not entirely clear why the authors have used this approach, since the presented data only shows short-term intravital microscopy experiments and not multiple day experiments. Moreover, if the authors only want to show the spontaneous firing of Ca²⁺ transients, it seems more relevant to transplant the iSKM bundles in (injured) muscles for example in the leg and surgically expose the muscle for intravital microscopy.

We thank the reviewer for this comment and agree that the intravital window chamber studies could have been done at more than two time points post-implantation (PI). We therefore performed additional window chamber experiments and imaged and quantified blood vessel density at PI days 3, 6, 9, 12, and 15. We observed that blood vessel density in the iSKM implants progressively increased and after PI day 9 was significantly higher in the bundle vs. non-bundle area. These findings are now shown in new Figs. 6b, c and described in lines 253-257 of the revised Results section.

We also agree with the reviewer's comment that native muscle is a more relevant transplantation model for iSKM bundles. We therefore implanted iSKM bundles into the tibialis anterior muscle of NSG mice and performed Ca²⁺ transient measurements and immunostaining 1 week PI. To improve our ability to optically record Ca²⁺ transients from the intact muscle-implanted bundles, we transduced the cells with lentivirus encoding a red-shifted fluorescent Ca²⁺ sensor R-GECO². Consistent with window chamber results, we found that implanted bundles survived, maintained structure, and fired twitch and tetanic Ca²⁺ transients in response to electrical stimulation. These findings are now shown in new Figs. 6j-l,

Supplementary Fig. 15, and Supplementary video 5, and are described in lines 268-279 of the revised Results section and further discussed in lines 346-350.

Together, these results from the two mouse implantation models suggest that *in vitro* engineered iSKM bundles have potential to survive, vascularize, and function *in vivo*.

3. The description of the transplantation and intravital experiments lack all details. How many mice have been imaged, how many spontaneous firing of Ca²⁺ transients have been observed, etc etc. This is important to judge what the efficiency is of functional fibers upon transplantation.

We apologize for lacking details of these experiments. For the data presented in the revised Figs. 6b,c, we intravitaly imaged and quantified vascularization at 5 different time points in a total of 16 implanted iSKM bundles in 12 mice. For calcium imaging *in vivo* (as well as *in vitro*), spontaneous Ca²⁺ transients in bundles were observed only sporadically in individual myotubes (an example shown in Supplementary video 4), which was difficult to reliably quantify using GCaMP6 imaging. We therefore imaged and quantified GCaMP6-reported twitch and tetanic Ca²⁺ transients in implanted bundles (total of 18 bundles in 10 mice) during application of electrical stimulation *ex vivo*. For these experiments, electrical stimulation was applied to excised dorsal skin with engrafted bundles and movies of electrically induced GCaMP6 flashing were recorded by a CCD camera (examples shown in Supplementary video 4 and Fig. 6h,i). Notably, electrically induced Ca²⁺ transients were observed in 93% of all *ex vivo* studied implanted bundles (18/20 bundles in window chambers and 8/8 bundles in TA muscle). The numbers of mice and implanted bundles used for intravital and *ex vivo* analysis are now clarified in the revised legend of Fig. 6.

4. The authors have imaged the iSKM on day 7, 14, and 21 post-implantation. Can the authors describe whether the transplanted tissue changes over time? Can the authors show images of the same transplanted tissue over multiple days?

We have now intravitaly imaged the same implanted bundles at days 3, 6, 9, 12, and 15 and have added representative bundle images from these time points in revised Fig. 6b. We did notice that the implanted bundles change over time *in vivo* by: 1) having gradually increased blood vessel density (revised Fig. 6c), 2) becoming more flat and less cylindrical (Fig. 6d), and 3) increasing transparency (Supplementary Fig. 14) that potentially resulted from fibrin degradation (due to lack of an anti-fibrinolytic agent present in culture media, but absent *in vivo*) and/or tissue remodeling by host cells.

Minor comments:

1. In video 4, a spontaneous firing of Ca²⁺ transient should be visible. Can the authors show with an arrow in the movie where the firing cells is?

We thank the reviewer for this suggestion. The flashing myotubes are now shown with an arrow in the revised Supplementary video 4.

Reviewer #3: We thank the reviewer for the very helpful comments that we have answered in detail. Reviewer's comments are cited in bold. The answers are in normal font. Any citations from the main or supplementary text are shown in quotes and italics. Other minor changes in the text are made to improve readability and satisfy formatting requirements of *Nature Communications*.

General comment:

Authors have already reported bioengineered human myobundles using human primary myoblasts (2014, 2015). This time, Authors used human ES cells (1 clone) and iPS cells (1 clone) instead of primary myoblasts. The 3D myobundles seem better than already reported 2D hiPS cells-derived skeletal muscle for in vitro models or biomaterials for medicine, but the protocol is a collection and modification of already reported protocols (GSK3i, PAX7 overexpression, 3D bundle, Window chamber etc.).

We thank the reviewer for this comment. Our manuscript is the first to demonstrate engineering of 3D contractile skeletal muscle tissues derived from human pluripotent stem cells, which opens doors to new studies of human muscle physiology and disease. While we previously reported generation of functional tissues from primary human myoblasts³, the development of the hPSC-based technology involved extensive optimization of multiple intermediate steps to ensure high yield and purity of myogenic progenitors and optimal 3D culture conditions leading to successful formation and functional maturation of engineered muscle. Furthermore, we have directly compared molecular and functional properties of human PSC-derived and primary engineered skeletal muscle tissues, and have first-time investigated the *in vivo* fate of engineered human skeletal muscles implanted in both mouse dorsal window chamber and native hindlimb muscle (revised Fig. 6, Supplementary Figs. 14, 15 and Videos 4 and 5). In addition to detailed studies with 2 hPSC clones shown in the original manuscript, we now demonstrate generation of functional iSKM bundles from 2 additional hPSC clones (1 hiPSC and 1 hESC clone, new Supplementary Fig. 11).

Major comments:

1. Figure 5. It is more informative to present the expression level of each gene in mature muscle tissue for comparison.

We agree with the reviewer and have performed additional qPCR analysis of all genes from Fig. 5 in both native human muscle and primary engineered human muscle. Results of this analysis are presented in the new Supplementary Fig. 12 and further described in lines 233-248 of the revised Results section.

2. In the text, Authors should mention that specific force of human or mouse myofibers are more than 100 mN/mm².

We thank the reviewer for the suggestion. We added reported values of specific forces for both fetal and adult human native muscle in lines 326-328 of the revised Discussion section.

Minor comments:

1. Which sub-unit of AChR was examined?

We used aBTX that binds to α -subunit of the nicotinic acetylcholine receptor (AChR) of neuromuscular junctions (<https://www.thermofisher.com/order/catalog/product/B13422>).

2. In Supplementary Figure 1, Tra-1-81 was not expressed on the surface of the cells.

We thank the reviewer for this comment. To confirm the membrane expression of Tra-1-81, we performed FACS analysis of live undifferentiated TRiPS cells and showed that over 90% of cells

expressed Tra-1-81. These results are now shown in revised Supplementary Fig. 1c and flow cytometry analysis is described in lines 395-401 of the revised Methods section.

Reviewer #4: We thank the reviewer for the very helpful comments that we have answered in detail. Reviewer's comments are cited in bold. The answers are in normal font. Any citations from the main or supplementary text are shown in quotes and italics. Other minor changes in the text are made to improve readability and satisfy formatting requirements of *Nature Communications*.

1. While these results are exciting, I feel that further quantification is required to document the extent of differentiation, and that comparison with adult skeletal muscle is vital. Specifically, at several points of the manuscript nice example images are shown, without giving the reader an understanding of how reproducible the findings are across each muscle and between muscles. For example, when it comes to muscle structure, the authors point out the appearance of z lines, a-band, etc. How much of the muscle had this type of appearance? The z-lines visible in Figure 3h appears to be a bit wobbly. Was this a general finding? Were t-tubules present at regular interval and density, and were the dimensions of these similar to those seen in adult muscle?

We thank the reviewer for this important comment. Comparisons of mRNA expression levels in iSKM bundles and native adult skeletal muscle are now shown in new Supplementary Fig. 12 and described in lines 233-248 of the revised Results section. Furthermore, we have performed quantification of TEM images and presented results in new Supplementary Fig. 8. We found that the large majority of cells in 4-week iSKM bundles contained registered sarcomeres with alternating A-bands and I-bands (92% of cells), electron-dense Z-lines (92% of cells), and distinct M-lines (86% of cells) and H-zones (88% of cells) located at the center of A-bands. At the level of individual sarcomeres, the majority exhibited distinct Z-lines (72.6%), M-lines (54.3%), and H-zones (68.2%). The average sarcomere length was $2.11 \pm 0.67 \mu\text{m}$, comparable to resting sarcomere lengths of native muscles⁴. We also updated Fig. 3h to a lower magnification image to show more sarcomeres and better represent what we observed. While some sarcomere had straight Z lines, most Z lines were not perfectly straight. T-tubules in iSKM myotubes were present at relatively low density with the diameters that varied from 30-300nm, which is different from native skeletal muscle (20-40nm)^{5,6}, and more like cardiac muscle (20-450nm)^{7,8}. Additional images of T-tubules are shown in revised Supplementary Fig. 9. The above findings are now described in lines 182-194 of the revised Results section.

2. The authors present force data with a comparison to literature values, but these experiments should be done in the present article to ensure that experimental conditions are comparable.

Thank you for this suggestion. We have now performed functional studies in primary human myobundles differentiated under same conditions as iSKM bundles. The obtained results are presented in new Supplementary Fig. 13 and described in lines 242-245 of the revised Results section. Additionally, gene expression data comparing iSKM bundles and primary myobundles are shown in new Supplementary Fig. 12 and described in lines 233-241.

3. In implanted muscles, how variable is the capillary network and characteristics of the Ca transient? How do these compare with native muscle?

To address any concerns about variability and reproducibility of the *in vivo* studies, we increased sample numbers and quantified blood vessel density (BVD, results presented in revised Fig. 6). All the implanted bundles showed consistent increase in BVD with time post-implantation. We did not see much variation in the capillary network density between implanted bundles as evidenced by the relatively narrow error bars. With respect to calcium transients, ~93% of the studied implanted bundles (18/20 in window chambers and 8/8 in TA muscle) exhibited Ca^{2+} transients with amplitudes that were reproducibly lower than those of age-matched *in vitro* controls (revised and new Figs. 6 i, l). There is no data available to let us compare the BVD and Ca^{2+} transients between the implanted bundles and native human muscle. However, since specific force of adult native muscles is over 20-fold higher than that of iSKM bundles (Discussion, lines 326-328), and expression levels of Ca^{2+} -handling genes are also higher in native muscles (new Supplementary Fig. 12), we expect that calcium transient amplitudes in iSKM bundles are significantly lower than those of native muscle tissues.

References

- 1 Chen, T. W. *et al.* Ultrasensitive fluorescent proteins for imaging neuronal activity. *Nature* **499**, 295-300, doi:10.1038/nature12354 (2013).
- 2 Zhao, Y. *et al.* An expanded palette of genetically encoded Ca²⁺(+) indicators. *Science* **333**, 1888-1891, doi:10.1126/science.1208592 (2011).
- 3 Madden, L., Juhas, M., Kraus, W. E., Truskey, G. A. & Bursac, N. Bioengineered human myobundles mimic clinical responses of skeletal muscle to drugs. *Elife* **4**, e04885, doi:10.7554/eLife.04885 (2015).
- 4 Cutts, A. The range of sarcomere lengths in the muscles of the human lower limb. *J Anat* **160**, 79-88 (1988).
- 5 Cully, T. R. *et al.* Human skeletal muscle plasmalemma alters its structure to change its Ca²⁺-handling following heavy-load resistance exercise. *Nat Commun* **8**, 14266, doi:10.1038/ncomms14266 (2017).
- 6 Sandow, A. Excitation-contraction coupling in skeletal muscle. *Pharmacol Rev* **17**, 265-320 (1965).
- 7 Soeller, C. & Cannell, M. B. Examination of the transverse tubular system in living cardiac rat myocytes by 2-photon microscopy and digital image-processing techniques. *Circ Res* **84**, 266-275 (1999).
- 8 Brette, F. & Orchard, C. Resurgence of cardiac t-tubule research. *Physiology (Bethesda)* **22**, 167-173, doi:10.1152/physiol.00005.2007 (2007).

Reviewers' Comments:

Reviewer #1:

Remarks to the Author:

The authors have added new data and addressed my concerns. Specifically, the authors performed a direction comparison between their iSKM bundles, primary human myobundles, and native muscle, which was a key characterization step not present in the original submission. I have only two minor comments:

Major Comments

None

Minor Comments

1. The Methods section does not describe the source or procedures for the primary myotubes or native muscle.
2. Some of the bar graphs use black and white patterns that can be difficult to differentiate. Colors would be more apparent.

Reviewer #2:

Remarks to the Author:

The authors have sufficiently answered all my concerns. I now recommend publication.

Reviewer #3:

Remarks to the Author:

This reviewer found that authors revised the manuscript according to the reviewer's previous comments, therefore the reviewer has no more comments on the revised manuscript.

Reviewer #4:

Remarks to the Author:

The authors have appropriately addressed my previous concerns, and have considerably expanded both the presented data and the discussion. I have no further comments.

We thank the reviewer 1 for the valuable feedback. Reviewer's comments are cited in bold. The answers are in normal font and revised text is cited in italics.

Reviewer #1

Minor Comments

1. The Methods section does not describe the source or procedures for the primary myotubes or native muscle.

The Methods are now revised to include preparation of primary cells and myobundles, as follows:

***“Engineering of primary human myobundles.** Native human skeletal muscle samples were obtained through standard needle biopsy or surgical waste from donors with informed consent under Duke University IRB approved protocols (Pro00048509 and Pro00012628). Muscle samples were minced and digested with 0.05% trypsin for 30 min at 37°C. Isolated cells were centrifuged to remove residual enzyme and resuspended in PMM, then preplated for 2 h to reduce fibroblast fraction. After pre-plating, cells were seeded onto to a Matrigel (BD Biosciences) coated flask and expanded by passaging upon reaching 70% confluence. At passage 3 or 4, cells were detached from the flask and used to fabricate primary myobundles as described for iSKM bundles.”*

2. Some of the bar graphs use black and white patterns that can be difficult to differentiate. Colors would be more apparent.

We thank the reviewer for this comment. The bar graphs in the main figure 4 and supplementary figure 10 are now presented using different colors.